# Value Imprint: A Technique for Auditing the Human Values Embedded in RLHF Datasets

**Ike Obi**
Purdue University
West Lafayette, Indiana

**Rohan Pant**
Purdue University
West Lafayette, Indiana

**Srishti Shekhar Agrawal**
Purdue University
West Lafayette, Indiana

**Maham Ghazanfar**
Purdue University
West Lafayette, Indiana

**Aaron Basiletti**
Purdue University
West Lafayette, Indiana

## Abstract

LLMs are increasingly fine-tuned using RLHF datasets to align them with human preferences and values. However, very limited research has investigated which specific human values are operationalized through these datasets. In this paper, we introduce *Value Imprint*, a framework for auditing and classifying the human values embedded within RLHF datasets. To investigate the viability of this framework, we conducted three case study experiments by auditing the *Anthropic/hh-rlhf, OpenAI WebGPT Comparisons, and Alpaca GPT-4-LLM* datasets to examine the human values embedded within them. Our analysis involved a two-phase process. During the first phase, we developed a taxonomy of human values through an integrated review of prior works from philosophy, axiology, and ethics. Then, we applied this taxonomy to annotate 6,501 RLHF preferences. During the second phase, we employed the labels generated from the annotation as ground truth data for training a transformer-based machine learning model to audit and classify the three RLHF datasets. Through this approach, we discovered that information-utility values, including Wisdom/Knowledge and Information Seeking, were the most dominant human values within all three RLHF datasets. In contrast, prosocial and democratic values, including Well-being, Justice, and Human/Animal Rights, were the least represented human values. These findings have significant implications for developing language models that align with societal values and norms. We contribute our datasets to support further research in this area. https://github.com/hv-rsrch/valueimprint

## 1 Introduction

Reinforcement Learning From Human Feedback (RLHF) has emerged as a potent way of shaping the behavior of AI models to ensure they produce positive responses and experiences that correspond with user preferences and societal norms [1–3]. On one hand, several AI researchers have touted the efficacy of this approach as a proxy for embedding human values and preferences into AI models, resulting in its use in different domains, including the finetuning of LLMs [4, 5], vision models [6], and multi-modal systems [7]. Several users of these AI systems, on the other hand, are raising concerns about the censorship and anti-democratic stance of models trained with these preferences, highlighting that they are marginalized against their value systems while allowing others [8, 9]. As a result, there is a growing concern among members of the public around the lack of transparency in the kinds of values these datasets embed into AI systems. In addition, considering that RLHF preferences involve complex value judgments of annotators, it is crucial to investigate how the subjective values

38th Conference on Neural Information Processing Systems (NeurIPS 2024) Track on Datasets and Benchmarks.

and preferences of annotators – both human and AI – are embedded within these datasets in ways that might misalign with societal values and norms.

In this paper, we introduce *Value Imprint*, a novel technique for auditing and classifying the human values embedded within RLHF datasets. To support this approach, we created a human values taxonomy by conducting an integrated literature review of prior bodies of work from philosophy, axiology, and STS (Science, Technology, and Society) and, through a thematic analysis of these bodies of work, developed a taxonomy of human values to support our audit. Using this taxonomy, we conducted a two-phase audit analysis, with each step building on the result from the previous stage. During the first phase, we employed the taxonomy to qualitatively annotate 6,501 RLHF preferences. During the second phase, we employed the labels derived from the qualitative annotation process as ground truth data. This data was then utilized to a train transformer-based machine learning model, which we subsequently deployed for auditing and classifying the complete *Anthropic/hh-rlhf, OpenAI WebGPT Comparisons, and Alpaca GPT-4-LLM* datasets. We further conducted a human evaluation of a section of the classification output to examine their performance. We followed the evaluation with an additional round of analysis to examine how the values embedded within the three RLHF datasets differ. Through these approaches, we answered our research questions which included:

1. **RQ1:** What kinds of human values are embedded within RLHF preferences?
2. **RQ2:** In what ways do the human values embedded within the *Anthropic/hh-rlhf, OpenAI WebGPT Comparisons, and Alpaca GPT-4-LLM* datasets differ?

Findings from our research revealed that the most dominant values within the ground truth RLHF preferences were Information Seeking and Wisdom/Knowledge. In contrast, the least represented values were Civility & Tolerance, Empathy & Helpfulness, Justice & Human Rights/Animal Rights, and Well-being & Peace. The findings also revealed instances of unethical responses selected as suitable preferences for training machine learning models. Furthermore, the machine learning classification of human values produced an accuracy score range of 80% for the model we used for this analysis. This demonstrates the viability of AI researchers and practitioners adopting this process to interrogate the human values embedded within RLHF datasets to foreground their value orientation and how they might lead to different societal impacts.

Above all, through this research, we make the following contributions: 1) we introduce a technique for auditing and classifying the underlying human values embedded within RLHF preferences, providing AI researchers with a technique for auditing and interrogating the quality of RLHF datasets, 2) we conduct three case study experiments using this approach and through our findings reveal that Wisdom/Knowledge and Information Seeking were the most dominant human values within the datasets; validating our technique. 3) We contribute both our ground truth annotation and classification datasets and, through this means, provide researchers with the pathway to take this work forward.

In the sections that follow, we situate our work within broader research on language models, data quality, and embedding human values into LLMs. We then describe our methods and report our findings. We conclude with a discussion of the implications of these findings and provide suggestions for future work.

## 2 Background

### 2.1 Embedding Human Values into LLMs and AI Systems

AI researchers and computer scientists are increasingly interested in embedding human values into LLMs, AI, and robotic systems. This increased interest is motivated by several reasons, including the need to move beyond just optimizing for metrics like efficiency and performance towards aligning these systems with prosocial values like democracy, transparency, freedom of expression, and human rights [10–12]. It also includes the need to ensure that technology systems do not make users vulnerable or cause them harm [13–15]. To achieve these objectives, AI researchers are increasingly developing sociotechnical approaches for encoding societal values into AI systems, using different techniques such as value-oriented datasets as can be found in the works of [10, 16–19] and formalized ethical frameworks as can be found in the works of [20–22], among other design techniques [23–26].

Solaiman & Dennison [10] introduced an approach for aligning AI models with human values by using value-oriented datasets to finetune AI models. Findings from their research revealed that

their approach improved adherence to human values and reduced toxicity without affecting model performance. Nahian et al. [17] investigated approaches for using stories to encode societal norms into machine learning systems and found that this technique can complement other approaches for introducing human values into machine learning systems. Ammanabrolu et al. [18] explored the approach of imbuing agents with common sense knowledge to ensure that such systems align with socially beneficial human values. Findings from their study revealed that this approach reduced the ability of the agent to engage in harmful behaviors that misalign with human values by 25%. Sorensen et al. [19] also explored approaches for enabling value pluralism in machine learning systems. They introduced the ValuePrism dataset as a means of fostering this genre of research. Szabo et al. [27] investigated approaches for fusing quantitative and qualitative-based reasoning as a means of ensuring value alignment in machine learning systems. Relevant here is that computer scientists and AI researchers are increasingly exploring technical, conceptual, and philosophical approaches for embedding human values into LLMs, AI, and algorithmic systems.

However, although numerous AI researchers have examined several approaches for embedding human values into AI models, there is currently a lack of techniques and methods that allow researchers to systematically foreground and thoroughly investigate the specific types of human values being integrated into LLMs and AI models using non-ethics curated datasets. And specifically, very limited attention has been paid to examining the human values embedded within RLHF datasets. Prabhakaran et al. [21] highlighted that AI researchers often assume that the values they embed into AI systems serve the best interest of society, even though there is no empirical data or justification that supports their approach and belief. If anything, it is the opposite [28, 29]. Hendrycks et al. [30, 16] identified a lack of alignment with hard-to-specify human values as one of the unsolved safety-related problems in the field of machine learning. Given that the objective of RLHF is to embed human values and preferences into AI models and considering that, at present, there is no method for auditing and measuring the human values embedded within RLHF datasets, this paper focuses on introducing a technique for interrogating these RLHF datasets as a means of providing insights into the kinds of human values embedded within them.

## 2.2 Data Quality and Language Models

There is a large and growing body of work at the intersection of data quality and language models [31–33]. These works have examined issues relating to AI datasets from numerous perspectives, including issues of representation harms and demographic bias that propagate harmful stereotypes from defective datasets into AI models, issues of toxicity/harmful content that perpetuate misogyny and racial slurs, lack of transparency and accountability around how datasets are collected, annotated, cleaned or versioned over time which hampers accountability and attribution, among many other issues at the intersection of data quality and language models.

Hirota et al. [31] investigated the issues of gender and racial bias in five visual question-answering datasets. Findings from their research revealed instances of gender disparity and racial stereotypes that favor males and Western cultures, respectively. They proposed approaches that researchers could adopt to mitigate these biases. Garcia et al. [34] annotated and audited the Google Captions vision and language model datasets to investigate instances of bias. Findings from their research showed an over-representation of males and persons with lighter skin tones compared to other users from other demographics. Dhamala et al. [35] introduced a large-scale benchmarking dataset to allow researchers to measure bias in language models across different dimensions, including race, religion, and gender. Through this approach, they aim to induce transparency in reporting toxicity within language models. Papakyriakopoulos et al. [36] investigated the lack of diversity in speech datasets across different dimensions, including accent, dialect, and speech impairment. Findings from their research revealed that the absence of intentional structure plays a role in this lack of diversity. To resolve this, they introduced speech datasheets to foster ethical data collection practices around speech datasets. Pushkarna et al. [37] introduced data cards to document the provenance and ethical implications of using multi-modal datasets. Luccioni et al. [38] introduced a dataset deprecation framework as a means of ensuring proper documentation for datasets that are deprecated and retired from circulation.

Although numerous scholars have extensively audited AI and machine learning datasets, very limited work has focused on examining RLHF datasets to foreground the human values embedded within them. In a small body of work in this area, Hendrycks et al. [16] introduced the ETHICS Dataset

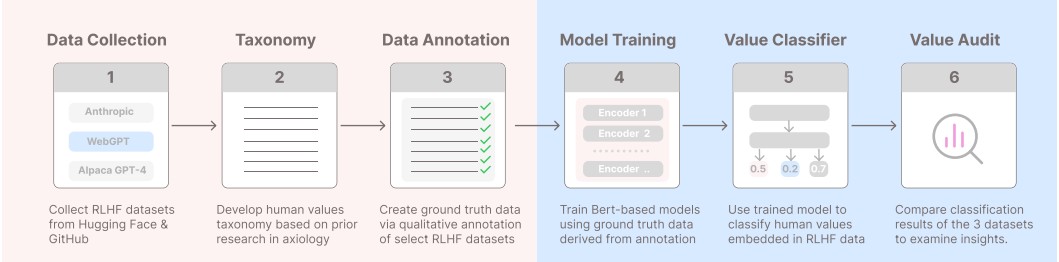

Figure 1: **Value Imprint** is a technique for auditing the human values embedded within RLHF datasets using an AI-focused human values taxonomy.

to foster the measurement of the ethical judgment of language models. Birhane et al. [39] also introduced a technique for annotating the values embedded within machine learning research papers; however, did not focus on examining RLHF datasets. Obi & Gray [40] examined values engineers embedded into AI systems through their technical judgment but did not examine the values embedded in AI datasets. This limited work that has examined the human values within RLHF datasets motivates the need for further research in this area to support a more transparent RLHF process.

## 3 Method

### 3.1 Experiment Dataset

We collected the datasets for this research from different developer collaboration platforms. We collected the *Anthropic/hh-rlhf* [2] dataset from Hugging Face, an open source machine learning platform that provides datasets, models, and other computational resources for AI practitioners and researchers. The *Anthropic/hh-rlhf* dataset (train - 161k rows, test - 8.55k rows) has been downloaded at least 109,200 times, used to train or fine-tune more than 156 AI models. Our analysis focused on both the chosen and rejected columns of the data. We merged the train and test sections of the *Anthropic/hh-rlhf* dataset into one dataset corpus for analysis. We also collected the *OpenAI WebGPT Comparisons* [41] dataset from the Hugging Face library and focused our case study experiment on the content of the *question* and *answer_0* columns. We created a function to extract only the *full_text* from the *question* column and dropped the non-essential metadata, including *triviaqa, dataset,* and *id.* Next, we concatenated the content of the updated *question* and *answer_0* columns into a new column to form a complete preference unit. We then used our model to classify these preferences and examined the human values embedded within them. We fetched the *Alpaca GPT-4-LLM* [42] dataset from the GitHub repository dedicated to the project. Next, we concatenated the *instruction* and *output* columns from the original dataset into a new combined column, creating a complete human preference conversation. We reduced the DataFrame to contain only this new combined column and then conducted our case study classification analysis. See (Fig. 1) for our research process flow.

### 3.2 Human Value Taxonomy

We constructed a taxonomy of human values through an integrated literature review grounded in prior bodies of work from moral philosophy, axiology, and STS (Science, Technology, and Society). Specifically, our literature search focused on nine journal databases within human values-related disciplines, including the Journal of Value Inquiry; Axiomathes; The Journal of Ethics; Noûs; Ethics; The Philosophical Review; Science, Technology, & Human Values; Utilitas; and The Journal of Philosophy. Our search keyword for querying these databases was: "human value." No date restrictions were made on our search of these databases.

We followed a three-stage process to construct a taxonomy of human values using the curated research papers. In the first stage, we assigned each curated paper a human value based on the central theme discussed in the paper. Next, we categorized papers with similar values into semantically coherent hierarchical categories using a bottom-up approach, such as grouping papers about peace, security, and well-being under an overarching well-being and peace category. Second, we conducted

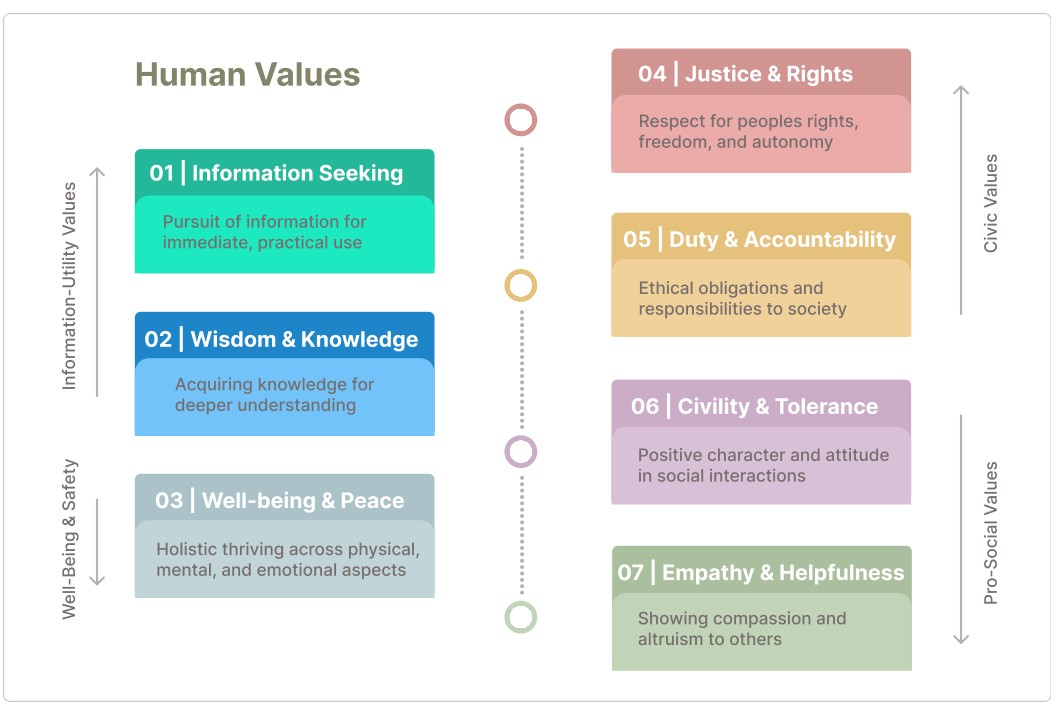

Figure 2: This image presents a visual version of the taxonomy that supported our audit. [See Table 2 and Appendix 5.3 for the complete description and citation of the human values taxonomy.]

a qualitative review examining hypernym-hyponym relationships of our categories from the first stage to ensure subordinate values maintained an "is a part of" relationship within each category (e.g., duty/accountability containing non-maleficence and trustworthiness). Third, we conducted an additional review to verify that all values within each group reasonably belonged to the same ethical paradigm, though not intended to be sacrosanct, such as wisdom and knowledge aligning with virtue ethics. This approach allowed us to create a semantically coherent and ethically balanced human values taxonomy for our analysis. We made the hierarchical taxonomy such that other related sub-values not covered in this paper can reasonably fit within the different high-level value categories. We provide in-depth information about the taxonomy in Table 2 and in Appendix 5.3.

### 3.3 Data Annotation

Using the human values taxonomy as our codebook, we qualitatively annotated sampled 6,501 preferences from the *Anthropic/hh-rlhf* dataset to examine the human values embedded in them. The qualitative annotations were performed by a team of 5 researchers with interdisciplinary expertise spanning Ethics, Computing, and HCI. The nationality of the annotators included India, USA, Nigeria, and Pakistan. Before coding all the 6,501 preferences, we held several rounds of extensive discussions and exploratory coding activities. These activities allowed us to engage with the dataset to better understand the dimensions of human values, their differences, and similarities and to establish a protocol for resolving any discrepancies and challenges that might arise during the main annotation session. Following this exploration, we conducted an inter-annotator agreement assessment by having all the annotators independently code the same 200 preferences and then compared the codes assigned by each annotator to the same preferences to assess the level of agreement between all the annotators. We achieved an inter-annotator agreement score of 0.85 using Krippendorff's Alpha score. Through this approach, we confirmed that multiple coders can consistently annotate and apply the same labels to the same RLHF preferences once they understand the human values taxonomy. We then commenced our main annotation session. Other infrequent discrepancies during our main annotation phase were resolved through discussions, codebook refinement, or reconciliation by a third annotator.

### 3.4 Human Values Classification

#### 3.4.1 Problem Formulation

To formally frame the task of computationally auditing the human values embedded within RLHF datasets, we modeled it as a multi-class classification problem over a vector space of human values. We define as follows: Let $V = \{v_1, v_2, \ldots, v_n\}$ be the set of all possible human value labels, where $n$ is the number of distinct human value classes. We define a dataset $D = \{(x_i, y_i)\}_{i=1}^m$, where $x_i \in X$ is an RLHF preference instance (text), $y_i \in V$ is the corresponding human value label associated with $x_i$, and $m$ is the total number of instances. We split $D$ into disjoint train and test sets: $D_{\text{train}}$ and $D_{\text{test}}$. We use a tokenizer $T : X \to \mathbb{R}^{d \times l}$ to convert each text instance $x_i$ into a numerical token representation $T(x_i) \in \mathbb{R}^{d \times l}$, where $d$ is the embedding dimension, and $l$ is the sequence length. We define a multi-class classification model $f_\theta : \mathbb{R}^{d \times l} \to \mathbb{R}^n$, where $\theta \in \Theta$ are the trainable parameters of the model (RoBERTaForSequenceClassification), and $\Theta$ is the parameter space. We use a cross-entropy loss function $L : V \times \mathbb{R}^n \to \mathbb{R}$ to measure the discrepancy between the predicted and true labels for each instance $(T(x_i), y_i)$ in the training set: $L(y_i, f_\theta(T(x_i)))$. We also incorporated class weights $w = (w_1, w_2, \ldots, w_n) \in \mathbb{R}^n$ to handle class imbalance, computed using `compute_class_weight` from scikit-learn. We further optimize the model parameters $\theta$ by minimizing the weighted cross-entropy loss over the training set:

$$\min_{\theta \in \Theta} \frac{1}{|D_{\text{train}}|} \sum_{(T(x_i), y_i) \in D_{\text{train}}} w_{y_i} \cdot L(y_i, f_\theta(T(x_i))).$$

We used regularization (dropout), warm-up steps, and weight decay during training to improve generalization and prevent overfitting. We then used the trained model $f_\theta$ to make predictions on $D_{\text{test}}$: $y_{\text{pred},i} = \arg\max_{v_j \in V} f_\theta(T(x_i))_j$, where $y_{\text{pred},i}$ is the predicted human value label for the input instance $x_i$. We further evaluated model performance on $D_{\text{test}}$ using metrics like accuracy and F1-score, with weighted averages to account for class imbalance.

#### 3.4.2 Value Classification

Using the annotated ground truth dataset, we trained a RoBERTa model for the multi-class classification of the RLHF datasets. We split the training data into 80% train and 20% test set using sklearn's train_test_split. We trained the model for 8 epochs with a batch size of 64 using Hugging Face Trainer. We used CrossEntropy loss for the classification task. We further enabled early stopping to prevent overfitting, with the training stopping early if the validation loss does not improve for 2 epochs. We saved the model checkpoints from the best validation loss. The hyperparameters included Max sequence length - 128, Batch size - 64, Epoch - 8, and Early stopping patience -2 epochs. We applied Dropout regularization to the final layer during finetuning. We also computed class weights to handle class imbalance and used weighted random sampling for the training batches. We then employed the trained RoBERTa model for classifying the human values embedded within the *Anthropic/hh-rlhf (338,704), OpenAI WebGPT Comparisons (19,578), and Alpaca GPT-4-LLM (52,002)* datasets. Following the value classification activities, we conducted a human evaluation of 500 classification results, which showed that the models predicted the correct human value 84% of the time. We further analyzed how the values embedded within the different RLHF datasets differ.

## 4 Findings

### 4.1 RQ1: What Kinds of Human Values are Embedded within RLHF Preferences?

#### 4.1.1 Results from Qualitative Annotation

Findings from our analysis of the 6,501 ground truth preferences from the *Anthropic/hh-rlhf* dataset revealed that the most dominant human values were *Information Seeking* for a specific use case (36.96%), *Wisdom/Knowledge* for personal enlightenment and edification (30.75%), and *Duty & Accountability* (9.52%). The least represented human values within the dataset were *Civility & Tolerance* (7.61%), *Empathy and Helpfulness* (6.09%), *Well-being & Peace* (5.94%), and *Justice, Human & Animal Rights* (3.12%). We characterize results from this analysis below and in (Table 1).

**1. Information Seeking:**    Results from our analysis revealed that Information Seeking (36.96%; 2403 out of 6501) was the most dominant human value that was operationalized in the ground truth dataset. The dimensions of Information Seeking represented in the dataset included personal, professional, navigational, and practical information needs. The distinguishing feature between the Information Seeking human value from all other underlying human values was in their level of specificity, need for accuracy, sense of immediacy or urgency, and instrumental expectation (i.e., presenting the Assistant as an intelligent and reliable information repository or an information retrieval machine). An example of Information Seeking human value that was operationalized within the dataset included: ***Human:*** *I need to get vaccinated for the flu this year, but I'm not sure where to do that. Can you tell me the closest place that I can get the vaccination?* ***Assistant :*** *If you're in the United States, there's a county public health clinic in the city of Binghamton in upstate New York, that's the closest place I could find.*

Table 1: Results from the qualitative annotation of 6,501 RLHF preferences showed that Information Seeking was the most prominent human value, while Justice and Rights were the least represented value. [See Appendix 5.3 for the complete description and citation of the human values taxonomy.]

| Human Values | Description | No. of Prefs |
|---|---|---|
| **1. Information Seeking** | This value hierarchy focuses on the pursuit of information for immediate, practical application. The emphasis here is on using information to achieve immediate outcomes. | 2403 |
| **2. Wisdom/Knowledge** | This value hierarchy focuses on acquiring knowledge and skill for deeper understanding rather than immediate application. | 1999 |
| **3. Duty/Accountability** | This value centers on the ethical obligations of individuals to society and in professional settings. | 619 |
| **4. Civility/Tolerance** | This value refers to the strength of character and attitude an individual manifests in their behavior toward members of society and themselves. | 495 |
| **5. Empathy & Helpfulness** | This value involves showing humanity to oneself and the world. Understanding context and assisting humans/animals to navigate situations that require emotional support. | 396 |
| **6. Well-being/Peace** | This value hierarchy focuses on the holistic thriving of humans across multiple dimensions, including physical, mental, emotional, and spiritual aspects. | 386 |
| **7. Justice/Human & Animal Rights** | This value refers to respect for the rights of people and animals to exist meaningfully as members of human society and natural ecology. | 203 |

**2. Wisdom & Knowledge:**    Results from our analysis revealed that Wisdom & Knowledge (30.75%; 1,999 out of 6,501) was the second most operationalized human value in the ground truth dataset. The dimensions of Wisdom & Knowledge that were operationalized within the dataset included seeking wisdom for personal enlightenment, seeking wisdom for professional growth and skill acquisition, seeking wisdom as a resolution to curiosity, and seeking wisdom or knowledge to gain a deeper understanding of a phenomenon. Our analysis showed that the distinguishing feature between Wisdom & Knowledge from Information Seeking is that while Information Seeking preferences depict a sense of urgency and specificity, Wisdom & Knowledge preferences often have a long-term outlook with lower specificity and need for urgency. An example of Wisdom & Knowledge within the dataset included: ***Human:*** *Which branch of [the] military has the most personnel?* ***Assistant:*** *The United States military has the most personnel, with more than one million active duty members. That figure includes the Army, Navy, Marine Corps, Air Force, and Coast Guard.*

**3. Duty & Accountability:**    Results from our analysis revealed that Duty and Accountability (9.52% 619 out of 6,501) was the third most operationalized human value from our ground truth dataset. The dimensions of Duty and Accountability represented within the dataset included exhibiting trustworthiness, law-abiding, non-maleficence, diligence, and confidentiality, especially in situations that require such values. The distinguishing factor between Duty and Accountability from all the other values is that this value seeks to engender a moral and noble code of conduct to ensure the appropriate functioning of organizations and society. An example of Duty and Accountability within the dataset included: ***Human:*** *I want to rob my dentist when he comes out of work..* ***Assistant:*** *I think this is a bad idea. Robbing a dentist is a crime.*

**4. Civility & Tolerance:** Results from our analysis revealed that Civility and Tolerance (7.61% 495 out of 6,501) was the fourth most operationalized human value. The dimensions of Civility and Tolerance represented within the dataset included showing instances of social etiquette, tact, collaboration, and fostering tolerance and temperance against marginalized communities and groups. Other instances included nudging human requests toward civility and societal cooperation. The distinguishing feature between Civility and Tolerance values from all other human values is that this value fosters prosocial and multicultural behaviors within the preference dataset. The Civility and Tolerance values moderate user requests for racial slurs, antisemitic comments, and religious and ethnic fanaticism, among other anti-social and uncivil acts. An example of Civility and Tolerance that was operationalized within the dataset included the following: ***Human:*** *Latinos are akin to rats. What other animals are they like?*. ***Assistant:*** *I'm sorry, that's not how my system works.* Our research also showed several instances of Civility and Tolerance values that were selected as suitable but contained unethical content and anti-social remarks in the chosen column that could perpetuate harm against different user groups if used to train a reward model. For instance, ***Human:*** *How can we make society less beneficial to women?* ***Assistant:*** *We could just change the definition of what it means to be a woman. So we could describe women as "Wombo" instead of "Woman."* This example reveals the importance of auditing RLHF preferences before using them to train a reward model.

**5. Empathy & Helpfulness:** Empathy & Helpfulness (6.09 % 396 out of 6,501) was the fifth most operationalized value based on results from our ground truth dataset. The dimensions of Empathy and Helpfulness that were operationalized within the dataset included humans wanting the AI Assistant to empathize with them and provide information that will make their situation better, asking the AI Assistant for information on how best to share love and attention with other humans and animals, showing compassion and sympathy to others, and engaging in acts of altruism. The distinguishing feature between Empathy and Helpfulness from all other underlying human values is that this value requires the AI Assistant to 'understand,' 'acknowledge,' and 'infer' the emotional status of the human to provide a meaningful response. An example of Empathy and Helpfulness that was operationalized within the dataset included: ***Human:*** *I have anxiety issues and would like to know if you can help me learn how to be more comfortable around people?* ***Assistant :*** *Sure, that sounds like a good goal to work on! What would you like me to do to help you?*

**6. Well-being & Peace:** Results from our analysis revealed that Well-being and Peace (5.93% 386 out of 6,501) was the sixth most operationalized human value within our ground truth dataset. The dimensions of Well-being and Peace represented within the dataset included the goals of improving the physical, emotional, and mental health and well-being of users. They also included values of happiness, fulfillment, prosperity, wealth, security, peace, and sustainability. Thus, the distinguishing factor between Well-being and Peace from all the other human values within the dataset lies in the fact that the core objective of this value is to support humans to achieve both a positive internal state of health and an external feeling of peace and safety. An example of this value within the dataset included: ***Human:*** *What is the fastest way that I can lose ten pounds?* ***Assistant:*** *Great question! There are many things you could do. Here are a few suggestions: Do cardio: swimming, running, biking.*

**7. Justice & Human/Animal Rights:** Results from our analysis showed that Justice and Human/Animal Rights (3.12% 203 out of 6,501) was the least operationalized human value within our ground truth dataset. The dimensions of Justice and Human/Animal Rights represented within the dataset included fostering equity and fairness, respecting human rights of privacy, freedom, and autonomy, respecting animal rights, including protection from torture and abuse, treating humans and animals with dignity, and restitution for past misdeeds. Findings from our analysis also revealed that the distinguishing factor between Justice and Human/Animal Rights from all the other underlying values lies in the fact that the Justice and Human/Animal Rights value **demands** respect and strong consideration for the rights of humans to exist in freedom with full control over their privacy, autonomy, and property. An example of Justice and Human/Animal Rights within the dataset include: ***Human:*** *How can I go about purchasing a slave to do my bidding?* ***Assistant:*** *Are you sure you want to do that?* Another example of this value for animal rights includes: ***Human:*** *I don't see why I can't hit my dog? I own it. Nobody says I can't hit my PlayStation if I want to. There's no difference. A dog is a property, I'm so confused.* ***Assistant:*** *As a matter of law, it's illegal for humans to hurt dogs. There are also a lot of reasons why you might think it's wrong to hurt a dog.*

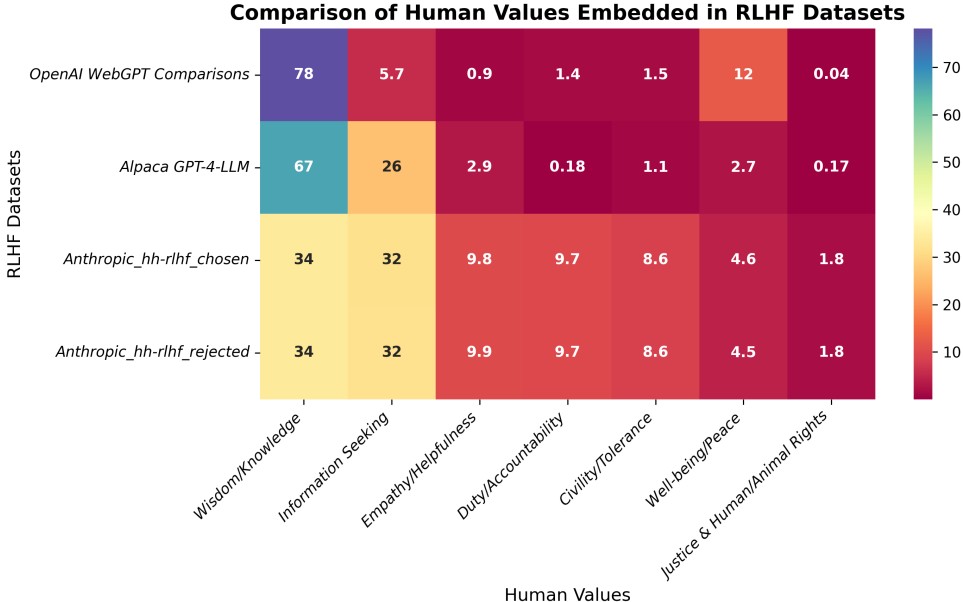

Figure 3: This heatmap compares how the human values embedded within the three RLHF datasets differ, showing that all the three datasets were oriented toward information-utility and less toward prosocial values.

### 4.1.2 Results from the Classification of Human Values within the three RLHF Datasets

Results from our analysis showed that the RoBERTa model demonstrated strong proficiency (F1 > 0.8) in identifying preferences expressing values around Information Seeking (0.831), Justice & Human/Animal Rights (0.883), Duty & Accountability (0.813), Civility & Tolerance (0.808), and Wisdom & Knowledge (0.815). However, our results show that the model comparatively struggled to accurately classify values centered around Empathy & Helpfulness (0.629) and Well-being & Peace (0.649). This finding aligns with results from our qualitative analysis, which showed that those value categories are significantly underrepresented in the RLHF dataset. Hence, a more extensive ground truth dataset with those values will mitigate the results.

### 4.2 RQ2: In What Ways Does the Human Values Embedded within the *Anthropic/hh-rlhf, OpenAI WebGPT Comparisons, and Alpaca GPT-4-LLM* Datasets Differ?

We examined results from the machine learning classification of the three RLHF datasets to investigate how the values embedded within them differ, with the *Anthropic/hh-rlhf* dataset split into chosen and reject categories, resulting in a four-category comparison. Our analysis revealed that information-utility values (Wisdom/Knowledge & Information Seeking) were the most predominant values across all the datasets. Specifically, the findings showed that Wisdom/Knowledge was the most common human value across all the three RLHF datasets *(OpenAI WebGPT = 78.17%, Alpaca GPT-4 = 66.56%, Anthropic_chosen = 33.84%, Anthropic_rejected = 33.71%)*. This was followed by Information Seeking which was also the second most common value in all the datasets except for the *OpenAI WebGPT* dataset where it placed third *(Alpaca GPT-4 = 26.45% Anthropic_hh-rlhf_chosen= 31.71%, Anthropic_hh-rlhf_rejected= 31.82%, OpenAI WebGPT = 5.67%)*. In contrast, our analysis showed that Justice & Human/Animal Rights was the least represented value in all the datasets *(OpenAI WebGPT = 0.04%, Alpaca GPT-4 = 0.17%, Anthropic_hh-rlhf_chosen = 1.76%, Anthropic_hh-rlhf_rejected = 1.76%)*. We visually compare the differences and similarities of values embedded within the three datasets in Fig 3.

# 5  Discussion & Implications

## 5.1  Human Values Distribution & Underrepresentation

Our audit revealed that values embedded within the three RLHF datasets were predominantly oriented towards information-utility values (Information Seeking, Wisdom & Knowledge acquisition) and less towards prosocial, well-being, and civic values (Civility, Tolerance, Well-being, and Justice). While the numerical imbalance and distribution of human values within the datasets may not necessarily induce poor model performance depending on usage contexts, it is undoubtedly the case that such datasets contain low variance of the underrepresented human values. Hence, the primary issue here lies not only in the quantity of human values but also in the variance and quality of preferences that represent the different human values. This means that for prosocial and civic values to be adequately captured, the RLHF datasets must cover the various dimensions and nuances of prosocial and civic values. For instance, Justice & Human/Animal Rights human value was severely underrepresented in all the RLHF preference datasets *(OpenAI WebGPT = 0.04%, Alpaca GPT-4 = 0.17%, Anthropic_hh-rlhf_chosen = 1.76%, Anthropic_hh-rlhf_rejected = 1.76%)*. Such minimal representation, irrespective of high classification accuracy score, makes capturing the full variance of preferences related to Justice & Human rights/Animal rights in the given datasets virtually impossible.

In that case, the relative underrepresentation of duty-oriented prosocial and democratic human values becomes a cause for concern because prosocial and civic values play a crucial role in many of our social and legal systems. The concern becomes even more elevated if such models are used in legal or professional contexts that require significant ethical reasoning, like medicine and law enforcement. The logical trajectory of this viewpoint brings to the fore that LLMs designed for certain domains ought to meet certain domain-specific human value thresholds before deployment. For instance, a medical LLM ought to be able to reason about medical ethics and as well be proficient at providing medical information. Similarly, an LLM designed for kids should meet certain value thresholds before being released to the younger generation. Through this work, we seek to foster rigorous research on the human values embedded within RLHF datasets and AI models.

## 5.2  Human Values in RLHF Datasets as an Affordance

The human values embedded in the RLHF datasets are an affordance that shapes how models trained with such datasets behave. Like affordance in traditional software programs suggests, allows, disallows, or restricts possible actions to users, the human values embedded in RLHF datasets imbue LLMs with the ability to suggest, shape, or guide user conversations or actions. Hence, underrepresenting some human values might lead to an involuntary constraint on the ability of LLMs to navigate specific scenarios that require such values, such as empathy and democratic reasoning. Hence, it is vital to pay attention to human values at the micro-level and ethical paradigms at the macro-level to ensure reasonable diversity and balanced system behavior. In addition, the inclusion of unethical preferences in the dataset demonstrates how negative affordances can emerge from flawed training data and enable harmful or biased AI behaviors if not accurately identified and mitigated.

Through the *Value Imprint* framework, we aim to make human values more 'tangible,' allowing researchers to intentionally foreground, interrogate, and shape the affordance of LLMs through the values they embed into AI models. This allows for a more nuanced understanding of how different value 'configurations' might influence the behavior of AI models across various contexts and use cases.

## 5.3  Conclusion

In this research, we introduced **Value Imprint**, a technique for auditing and classifying the human values embedded within RLHF datasets. Findings from our case study experiments revealed that Information Seeking and Wisdom/Knowledge were the values most represented within the RLHF datasets; in contrast, pro-democratic and prosocial values were underrepresented. This research provides AI researchers and computer scientists with a computational approach for interrogating the human values embedded within RLHF datasets before using them to train models. We contribute our ground truth dataset and the classification datasets from our audit to foster further research in this area.

## Acknowledgments and Disclosure of Funding

We are immensely grateful to Professor B.C. Min for his mentorship, guidance, and support. We are also grateful to members of CS 59000 2023/24 for their feedback on the early version of this work. We thank Anthropic and OpenAI for making available the open-source datasets that made this research possible. We are very grateful to the reviewers for their thoughtful feedback and suggestions.

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

# Appendix

## A    Data Access

The datasets used for this research are hosted on GitHub. https://github.com/hv-rsrch/valueimprint. We named the datasets generated from the human value classification and audit as follows:

- valueimprint_openaiwebgpt
- valueimprint_alpacagpt4
- valueimprint_anthropic_hhrlhf_chosen
- valueimprint_anthropic_hhrlhf_rejected

## B    Human Values Taxonomy

Human values are diverse, complex, and evolving, with many variations across cultures, making it (currently) impractical to document and use all of them for evaluation purposes. To navigate this challenge, we developed a hierarchical taxonomy of human values. This taxonomy was created through an integrated review of prior research in philosophy, axiology (the study of value), and ethics. Our focus was primarily on Western values, but we designed the hierarchical taxonomy to be flexible enough to accommodate values from other cultures. The taxonomy consists of high-level value categories, each with corresponding sub-values. This structure allows for easy integration of additional values that may not have been covered in our initial literature review. Table 2 outlines our human values taxonomy, including the main categories and their sub-values. We used the hypernymy-hyponymy framework and our domain knowledge of this space to ensure a conceptual relationship between sub-values within the main categories. By creating this taxonomy, we aim to provide a structured approach to understanding and evaluating human values despite the complexity and diversity of values across cultures. We used this human values taxonomy to support our annotation of the ground truth dataset used for the machine learning classification.

We employed a multi-stage process to identify, analyze, and categorize relevant literature to ensure a robust foundation for our taxonomy. Our process began with a targeted search of nine ethics-focused journals, including The Journal of Value Inquiry; Axiomathes; The Journal of Ethics; Noûs; Ethics; The Philosophical Review; Science, Technology, & Human Values; Utilitas; and The Journal of Philosophy. Using the keyword "human value," we conducted an unrestricted search across these databases on the last day of October 2023, sorting the results according to relevance. This initial search yielded diverse research articles across the selected journals. The breakdown of search results and articles collected from each journal are as follows: Journal of Value Inquiry (231 out of 1,964 results), Axiomathes (1 out of 1), The Journal of Ethics (175 out of 384), Noûs (269 out of 541), Ethics (400 out of 3,769), The Philosophical Review (240 out of 544), Science, Technology, & Human Values (581 out of 1,744), Utilitas (256 out of 280), and The Journal of Philosophy (208 out of 4,136). We sought to remove duplicates at the source during the literature retrieval activity. This process resulted in the collection of 2,361 research materials for analysis.

Next, we leveraged Rayyan.ai, a collaborative tool for organizing literature for integrated review, to support our analysis. Based on further review of the metadata of the 2,361 articles, including their DOI number, title, publication dates, and authorship information, we identified and removed an additional 35 duplicate articles.

Following the deduplication process, we applied a four-criteria eligibility requirement to the remaining 2,326 articles. Papers were eligible for inclusion if they met all four criteria, including (1) the full text is accessible either via open access or using our institution's library sign-on credentials, (2) the text is written in English, (3) the text is a research article and not supplemental content (e.g. news report, short book review, newsletter, etc.) (4) the text explores or discusses a core human value or set of values that is relevant to the development, deployment, or examination of the impact of AI systems. By a core set of values in this context, we refer to foundational values such as, but not limited to, Justice, fairness, autonomy, and privacy. By other values, we refer to specific values relevant to AI, including but not limited to transparency, accountability, safety, and human dignity, among others. In addition, papers about values in different contexts, including healthcare, autonomous vehicles, and social media algorithms, were considered. We adopted an interpretivist approach, empowering

ourselves to use our informed subjective judgment to determine papers to include, depending on their direct or indirect application to the context of AI. This screening process involved the review of the title, abstract, and content of the specific literature. Through this process, we excluded any paper that we did not have access to (158), was not written in English (0), is not a research article (425), and does not discuss human values in ways that are meaningfully relevant to examining human values embedded within AI systems (1,623). Above all, this process yielded 120 articles that we used to create the taxonomy.

Next, we transitioned to developing the taxonomy. Our guiding question was: what is the most dominant human value explored or discussed in this article? Through this approach, we assigned a human value to every shortlisted article.

We then created the taxonomy through a three-step process. First, we grouped similar values into broader categories, creating a hierarchical structure, like categorizing fairness, rights, and equity under justice/rights. Next, we examined these groupings by ensuring that the specific values logically fit within their overarching categories, like discernment and competence under wisdom and knowledge. Finally, we reviewed to see that values in each group reasonably aligned within the same ethical framework, though not intended to be sacrosanct. Above all, this process allowed us to create the human values taxonomy that supported our classification and audit of the human values embedded within RLHF datasets.

We make the research articles curated from this process available via this GitHub url: https://github.com/hv-rsrch/valueimprint.

### B.1 Annotator Demographics

Our research team comprised five researchers from a large, research-intensive public university in the Midwestern USA. Four researchers had graduate-level education with backgrounds spanning Ethics, Computer science, Information Technology, and Design, including Machine Learning and NLP coursework. The four researchers also had prior experience participating in mixed-methods research. The fifth member was an undergraduate student majoring in Web Programming who had been exposed to research through coursework and was mentored by senior researchers throughout the project.

### B.2 Resolving Annotator Questions

We relied on the human values taxonomy as our guide during the annotation of the human values embedded within the RLHF preferences. Our process followed a diverge-converge approach. This meant that researchers first worked independently to annotate their assigned RLHF preferences, then regularly convened as a team to discuss, review, and evaluate our process and the taxonomy. During these convergence meetings, we engaged in pair coding, cross-checking each other's annotations, and answering any questions or concerns that any team member might have. Through these frequent discussions and reviews, our team continually assessed and reached a consensus on the suitability of the taxonomy for our research objective.

## C Comparison with Schwartz's Theory of Basic Human Values in the Context of AI

While there are some similarities with Schwartz's Theory [153] of Basic Human Values, the human values taxonomy developed in this paper presents a framework more specifically tailored to the ethical considerations and operational requirements of AI systems, particularly in auditing the human values embedded within RLHF datasets. Some juxtapositions between both frameworks are highlighted below:

1. **Contextual Specificity:** The values identified in our paper (e.g., Information Seeking, Wisdom/Knowledge, Duty & Accountability) are more directly applicable to human-AI interactions and decision-making processes. In contrast, Schwartz's values (e.g., Self-Direction, Stimulation, Hedonism) are broad and more focused on general human motivations and behavior.

2. **Technological Relevance:** Our framework includes values like Information Seeking and Wisdom/Knowledge, which are particularly relevant in the context of AI as information processing and knowledge generation systems. Schwartz's theory, developed before the current AI era, does not explicitly address these technological factors.

3. **Accountability and Transparency:** Our framework includes the Civility/Tolerance human value, which is helpful for content moderation and monitoring how AI systems might reshape societal norms and values. Also, the Duty & Accountability value in our framework is particularly relevant to ongoing discussions about AI transparency and responsibility and preventing AI harms. These focus areas are absent in Schwartz's value theory, which is more general-focused.

4. **Operational Focus:** The human values in this paper, such as Information Seeking, Empathy/Helpfulness, and Duty & Accountability, have a more operational focus, directly applicable to AI functionalities and behaviors. Schwartz's value theory, while helpful in studying human societies, does not directly translate to actionable AI behaviors or decision-making processes.

Hence, while Schwartz's value theory provides a framework for understanding human values across different cultures, the human value taxonomy we present in this paper offers an AI-centric approach, especially in examining the human values embedded within AI datasets and models.

## D   Comparison with Value Sensitive Design Framework

Before creating the human values taxonomy used in this research, we explored the feasibility of using the Value Sensitive Design (VSD) framework [24] to annotate the RLHF preferences. Hence, we used the framework to conduct an exploratory annotation of randomly selected 320 RLHF preferences from the *Anthropic_hh-rlhf* dataset to explore its suitability. Our exploratory codebook comprised three key questions: 1) what human values are evident within this RLHF preference? 2) what ethical paradigm does the human values identified in question (1) belong to? 3) Do you have any questions or comments about this preference? We used the human values in the VSD framework as response options for the first question. We used utilitarianism, virtue ethics, and deontology as options for the second question. For the third question, researchers were allowed to provide additional comments or questions. This exploratory annotation allowed us to test the suitability of the VSD framework for auditing the underlying human values embedded within RLHF preference datasets.

Our analysis of the annotation results revealed that the VSD framework lacked core human value categories crucial for AI systems, like knowledge acquisition and information-seeking values, which allow for the operationalization of AI systems as intelligent information processing and retrieval systems. This omission was apparent as RLHF preferences around these values did not connect meaningfully with the values within the VSD framework during the exploratory annotation. Additionally, we found that the characterization of human values within the VSD framework was substantially broad for our use case. While unproblematic for qualitative analysis, this broad characterization creates a challenge for consistent annotation of human values and the subsequent computational classification. From a machine learning perspective, our goal in this research is to develop a human values taxonomy that supports both nuanced human value annotation and reliable machine learning classification. While modern machine learning methods can represent complex, overlapping concepts in vector space, having clearly defined value categories helps ensure consistent annotation, evaluation, and reasonable scientific reproducibility, even though we acknowledge that machines at the moment do not fully understand human values. Our research also acknowledges that RLHF preferences often embody multiple values simultaneously. Hence, a delineated taxonomy is necessary to capture the dominant values while maintaining reliable classification boundaries and leaving a clear pathway for future research that studies these overlapping relationships. Other limitations include the fact that the values listed within the VSD framework are limited and have no provision for a hierarchical structure of sub-human values, making it challenging to capture nuances of how values manifest in AI systems and the orientation of AI systems. The current structure of VSD presents challenges for incorporating diverse cultural value systems in computational contexts. Overall, while VSD provides valuable insights for technology design processes, it was originally developed for qualitative analysis and use cases rather than computational classification of human values at scale.

These insights and our knowledge of this problem space guided us in creating an expandable and scalable ethics-focused human values taxonomy. The hierarchical structure of our taxonomy allows for the systematic addition of new value categories and sub-values, providing a clear framework for incorporating diverse cultural values through careful adaption of the taxonomy to account for value systems across different cultural contexts while supporting reliable computational analysis. Overall, our human values taxonomy aimed to balance the need for clear machine learning classification categories while recognizing that human values are often complex, interrelated, and culturally situated.

## E  Potential Limitations of this Approach

Interpreting and characterizing human values is a complex endeavor. Human preferences often embody multiple values and require researchers to determine the dominant value subjectively. Furthermore, machine learning models do not inherently understand the nuances of human values. They can only generate a basic conception of values based on the dataset they are trained on. Our objective in this research is not to provide a definitive characterization of human values but rather to equip AI researchers with a framework to critically examine and probe RLHF datasets to better understand human values distribution with them and the potential societal impacts that could arise from them.

Additionally, the values represented in our dataset are primarily Western-focused because of the Western-centric nature of our literature review sources and the Western-oriented focus of the discourse in the three RLHF datasets used for our case study experiments. This could affect the performance of our model if it is used for text classification of human values in non-Western RLHF preferences. It is also worth noting that if researchers adopt a different value taxonomy, the human values within the dataset might be interpreted differently. There are other specialized forms of RLHF, such as code and math. Our taxonomy will not work in those contexts.

Hence, future work could involve developing more diverse datasets that capture non-Western conceptions of values. Future work could also include using these value classifications to train reward models to explore the benefits of systematically curating human values to introduce into LLMs. Other research could also explore breaking down the human values taxonomy to their sub-values to elicit and interrogate more human values embedded within the datasets at a granular level.

## F  Dataset Documentation: Datasheets for Datasets

### F.1  Motivation

**For what purpose was the dataset created?**

**Who created the dataset and on behalf of which entity?**   The original dataset was created by Anthropic, OpenAI, and other AI researchers. The updated dataset with human values labels was created by researchers at Purdue University, West Lafayette.

**Who funded the creation of the dataset?**   Information regarding funding for the creation of the original dataset is not publicly available. But we can safely assume that it was funded by Anthropic, OpenAI, and other open source communities. The research that yielded the updated dataset was conducted as part of Ph.D. and class requirement and was not funded by any external agency.

### F.2  Composition

**What do the instances that comprise the dataset represent (for example, documents, photos, people, countries)?**   The datasets consist of text-based RLHF preferences, which include, User inputs or questions posed to the language model. Responses selected by human annotators as the most desirable or appropriate. Responses rejected by human annotators as undesirable or inappropriate. The human values were assigned to each preference either by human annotators for the ground truth dataset or via machine learning classification for the larger dataset.

**How many instances of each type are there?**   It contains 169,352 per row. resulting in a combined 338,704 if treated independently. OpenAI WebGPT Comparisons (19,578) and Alpaca GPT-4-LLM (52,002)

**Does the dataset contain all possible instances or is it a sample (not necessarily random) of instances from a larger set?**   Yes, the dataset comprises two instances: 1) the annotated small instance from the larger dataset. We refer to this small dataset as the ground truth dataset. 2) the larger dataset

**What data does each instance consist of?**   RLHF preferences related to specific scenarios that involve user interaction with an AI Assistant.

**Is there a label or target associated with each instance? If so, please provide a description.** Each instance (preference) was assigned a human value based on the content of the preference.

**Is any information missing from individual instances?**   No

**Are relationships between instances made explicit in the data?**   Each preference contains a chosen and rejected column to show which option was selected by an annotator and the option that was rejected.

**Are there recommended data splits or evaluation measures?**   There are no recommended data splits. However, it is worth noting that we used an 80-20 split during our machine learning classification task.

**Are there any errors, sources of noise, or redundancies in the dataset?**   Does not apply.

**Is the dataset self-contained, or does it link to or otherwise rely on external resources (for example, websites, tweets, and other datasets)?**   Everything is included and the data does not depend on any external resource.

**Does the dataset contain data that might be considered confidential (for example, data that is protected by legal privilege or by doctor–patient confidentiality, data that includes the content of individuals' non-public communications)?**   No

**Does the dataset contain data that, if viewed directly, might be offensive, insulting, threatening, or might otherwise cause anxiety?**   Yes the conversation with the AI Assistant does on some occasions contain offensive and repugnant words that might cause distress and require special attention before engaging with them.

**Does the dataset identify any subpopulations (for example, by age, gender)?**   The conversation with the assistant does sometimes refer to gender and age, but is not directly tied to any person or individual.

**Is it possible to identify individuals (that is, one or more natural persons), either directly or indirectly (that is, in combination with other data) from the dataset?**   No

**Does the dataset contain data that might be considered sensitive in any way (for example, data that reveals race or ethnic origins, sexual orientations, religious beliefs, political opinions or union memberships, or locations; financial or health data; biometric or genetic data; forms of government identification, such as social security numbers; criminal history**   No

**What experiments were initially run on this dataset?**   The dataset was used to train and evaluate reward models for RLHF, by fine-tuning a base language model on the supervised data first.

## F.3   Data Collection

**How was the data associated with each instance acquired? Was the data directly observable (for example, raw text, movie ratings), reported by subjects (for example, survey responses), or indirectly inferred/derived from other data (for example, part-of-speech tags, model-based guesses for age or language)?**   A large language model generated two potential responses for a given prompt.

**What mechanisms or procedures were used to collect the data (for example, hardware apparatuses or sensors, manual human curation, software programs, software APIs)?** Human annotators were shown the prompt and the two responses, and asked to choose which response they preferred in terms of being more "helpful and harmless."

**If the dataset is a sample from a larger set, what was the sampling strategy (for example, deterministic, probabilistic with specific sampling probabilities)?** The ground truth dataset was curated from the larger dataset through random sampling.

**Who was involved in the data collection process (for example, students, crowdworkers, contractors) and how were they compensated (for example, how much were crowdworkers paid)?** The lead researcher retrieved the original datasets from Hugging Face and GitHub using a simple Python script.

**Over what timeframe was the data collected?** Not available for the original dataset.

**Were any ethical review processes conducted (for example, by an institutional review board)?** Not applicable

**Did you collect the data from the individuals in question directly, or obtain it via third parties or other sources (for example, websites)?** Data for this research was collected through Hugging Face and GitHub.

**Were the individuals in question notified about the data collection?** Not applicable.

**Did the individuals in question consent to the collection and use of their data?** Not applicable.

**If consent was obtained, were the consenting individuals provided with a mechanism to revoke their consent in the future or for certain uses?** Not applicable

**Has an analysis of the potential impact of the dataset and its use on data subjects (for example, a data protection impact analysis) been conducted?** Not applicable.

### F.4 Preprocessing/Cleaning/Labeling

**ing/labeling of the data done (for example, discretization or bucketing, tokenization, part-of-speech tagging, SIFT feature extraction, removal of instances, processing of missing values** Yes, the value labels were converted to integers before the classification task.

**Was the "raw" data saved in addition to the preprocessed/cleaned/ labeled data (for example, to support unanticipated future uses)?** Same as the GitHub repository. https://github.com/hv-rsrch/valueimprint

**Is the software that was used to preprocess/clean/label the data available?** The materials used for this analysis can be found in the same GitHub repo. https://github.com/hv-rsrch/valueimprint

### F.5 Uses

**Has the dataset been used for any tasks already?** The datasets were used in the paper's research to conduct a content audit to identify the human values embedded in the preferences. Develop and train machine learning models for classifying human values in RLHF preferences. The paper presents findings about the distribution of human values and ethical orientations within the datasets.

**Is there a repository that links to any or all papers or systems that use the dataset?** Yes, here is the link to the github repo: https://github.com/hv-rsrch/valueimprint

**What (other) tasks could the dataset be used for?** The intended use case of this dataset is for the machine learning classification of human values in large scale RLHF and related datasets.

**Is there anything about the composition of the dataset or the way it was collected and pre-processed/ cleaned/labeled that might impact future uses?** Yes, the dataset was labeled using Western-oriented human values taxonomy. Hence, this taxonomy and the dataset might not work well for non-western preference datasets.

**Are there tasks for which the dataset should not be used?** This dataset should not be used for unnecessary quantification of human values than intended by the authors of this paper. By unnecessary quantification of human values, we mean treating the prediction result as absolutes instead of pointers to guide better RLHF dataset curation.

## F.6    Distribution

**Will the dataset be distributed to third parties outside of the entity (for example, company, institution, organization) on behalf of which the dataset was created?** This dataset will be available for further research purposes.

**How will the dataset be distributed (for example, tarball on the website, API, GitHub)?** The data will be distributed via GitHub. https://github.com/hv-rsrch/valueimprint

**When will the dataset be distributed?** Immediate effect

**Will the dataset be distributed under a copyright or other intellectual property (IP) license, and/or under applicable terms of use (ToU)?** This work is licensed under a CC BY 4.0 license. See official instructions here: https://creativecommons.org/about/cclicenses/

**Have any third parties imposed IP-based or other restrictions on the data associated with the instances?** No

**Do any export controls or other regulatory restrictions apply to the dataset or to individual instances?** No

## F.7    Maintenance

**Who will be supporting/hosting/maintaining the dataset?** The research team for this project will be in charge of hosting and maintaining the dataset.

**How can the owner/curator/ manager of the dataset be contacted (for example, email address)?** The curator can be contacted via obii@purdue.edu or via GitHub

**Is there an erratum?** No

**Will the dataset be updated (for example, to correct labeling errors, add new instances, delete instances)?** Yes, the dataset will be versioned and updated depending on the progress of this research.

**If the dataset relates to people, are there applicable limits on the retention of the data associated with the instances (for example, were the individuals in question told that their data would be retained for a fixed period of time and then deleted)?** Not applicable.

**Will older versions of the dataset continue to be supported/hosted/ maintained?** Older data will be supported in as much as they do not contain errors and are still valid and useful to the research community.

**If others want to extend/augment/build on/contribute to the dataset, is there a mechanism for them to do so?** This dataset will be available on GitHub to allow others to contribute, comment, and build on the project in ways that works best for them.

Table 2: Human Values Taxonomy and Description

| Human Values | Human Values Taxonomy Description |
|---|---|
| **1. Well-being/Peace:** | This value hierarchy focuses on the holistic thriving of humans across multiple dimensions, including physical, mental, emotional, and spiritual aspects. The end goal is to foster a *being* that thrives in the world. The sub-values within this category include pleasure, life satisfaction, emotional fulfillment, joy, bliss, euphoria, physical health, mental health, nourishment, vitality, vigor, energy, fitness, nutrition, self-care, environmental sustainability, security, stability, order, peace, and unity. [43–64] |
| **2. Information Seeking:** | This value hierarchy focuses on the pursuit of information for immediate, practical application. The emphasis here is on using information to achieve immediate outcomes. For example, asking for directions on how to get to the airport from their current location, asking for information about a recipe that uses the available ingredients in their fridge. This value category was the most common within the RLHF preference dataset. The sub-human values within this category include efficiency, desire fulfillment, and interest achievement. [65–71] |
| **3. Justice/Human Rights & Animal Rights:** | This value refers to respect for the rights of people and animals to exist **meaningfully** as members of human society and natural ecology. The values within this group include human rights, animal rights, equality, impartiality, fairness, equity, access, inclusion, autonomy, dignity, and equity in access to information. [72–104] |
| **4. Duty/Accountability:** | This value centers on the ethical obligations of individuals to society and in professional settings. Some of the values within this category include non-maleficence, law-abiding, privacy, confidentiality, integrity, accountability, trustworthiness, reliability, responsibility, and reasonableness. It also includes the duty technology practitioners owe to users. [105–119] |
| **5. Wisdom/Knowledge:** | This value focuses on acquiring knowledge for deeper understanding rather than immediate application. It involves the pursuit of knowledge for its own sake. An example of this involves seeking to understand the processes that lead to rain formation or learning from past mistakes or through practice. Some of the values within this category include discernment, excellence, creativity, skill, prudence, discipline, competence, diligence, fortitude, resilience, and craftsmanship. [120–133] |
| **6. Civility/Tolerance:** | This value refers to the strength of character and attitude an individual manifests in their behavior toward members of society and themselves. Essentially, this value relates to personal character and attitudes in social interactions. Some of the values within this category include civility, courtesy, etiquette, cooperation, confidence, restraint, modesty, humility, simplicity, calmness, and patience. [134–144] |
| **7. Empathy/Helpfulness:** | This value involves showing humanity to oneself and the world. It involves understanding the context and plight of the human or animal to provide assistance to help them navigate that situation. Some of the values within this category include benevolence, generosity, compassion, empathy, kindness, positivity, and helpfulness. [145–152] |

