# OpenReview forum: "Value Imprint: A Technique for Auditing the Human Values Embedded in RLHF Datasets"
_NeurIPS.cc/2024/Datasets_and_Benchmarks_Track — NeurIPS 2024 Track Datasets and Benchmarks Spotlight_

### Official Review · Reviewer_fZFM · 2024-07-21
**Review for RLHF Value Imprint**

**Rating:** 7
**Confidence:** 4

**Review:**

In short, this is a research and paper with a high potential impact and value, but for it to be acceptable at this venue, it needs much improved presentation/writing, and further clarifications about the taxonomy creation (Please see **Opportunities For Improvement**).

**Strengths:**

* a timely and important topic tackled in a creative way
* proper setup: taxonomy --> manual labels on a subset --> prediction on the rest of the dataset --> analysis
* results are insightful and call for further research
* the labelled Anthropic RLHF dataset can be useful for other researchers

**Additional Feedback:**

In my view, the paper can receive an accept, if the two main comments: 1) about the taxonomy and 2) thorough improvements of the writings are addressed.

**Clarity:**

No. That is the main issue with the paper.  Please see **Opportunities For Improvement** for how this reviewer believes the paper writing needs to be improved.

**Correctness:**

Overall, yes. Please see **Opportunities For Improvement** for how this reviewer believes the correctness can be better reported.

**Documentation:**

Yes, except for the taxonomy. Why the links/citations to the papers reviewed to form the taxonomy are not reported? It is not very clear how the values were extracted from papers -- was it so that each paper was discussing a single value? This whole process could be clarified much more.

**Ethics:**

OK.

**Limitations:**

Yes, but it would be good to discuss whether the RLHF instances could have been assigned multiple values and how the authors dealt with it. If not, why not?

**Opportunities For Improvement:**

* my main issue with this paper is how poorly it is written and presented (both the main text and appendix). there are numerous typos and inconsistencies in formatting. this really hinders the understanding and the trust that the authors can create a high quality manual taxonomy (i.e., a task demanding so much care) if the text cannot be carefully written and presented. Just to point to a few of the problematic examples in text:
     ** in abstract: "that wisdom/knowledge and information seeking were the most dominant human values within the RLHF dataset, while Justice, Human & Animal Rights were" --> wisdom/knowledge and information seeking is lowercase, while the other values are capitalised. Moreover, later in text wisdom/knowledge and information seeking are also capitalised.
   ** in Introduction, Lines 64-70: the formatting of the points 1) to 3) is totally inconsistent: some start capitalised, other do not; some are separate by ";" other by a "."
   ** in Appendix, there are instances of "It's" instead of "It is"
   ** throughout text, numbers are sometimes spelled without a comma (e.g., contained 338704 rows) and sometimes with it (e.g., n=2,890)
   ** Line 225: "Some examples of information-seeking preferences" and then only one example is given.
   ** Line 262: "I''m sorry"
* for the inter-annotator agreement among the 5 annotators, could you, please, clarify how it was calculated? how many options each annotator could annotate; and was it a single option per instance?
* for the review of literature to define the value taxonomy: how many articles were reviewed?
* Line 166 says:  "166 We first assigned each paper a human value." --> was it really always one value per paper? Which brings me to the next point:
* are the human values really mutually-exclusive? That is to say: whether each RLHF instance had only one value that could be assigned to it or did you sometimes have to chose the dominant value? This seems more likely to me and should be discussed both among the limitations as well as related to the annotation guidelines.
* The authors mention that they mostly focused on Western values: why is that, is it because of the literature review? Could you comment at all how your taxonomy relates to any other similar taxonomies of ethics frameworks, e.g., to Schwartz ten basic human values?

**Relation To Prior Work:**

The related work on human values, including those reviewed for the taxonomy, is missing.

**Summary And Contributions:**

This paper tackles a very important and interesting problem in a creative way: can we understand which values we are imprinting into LLMs by investigating the values of the preferences? To do so, the authors first conducted a literature review to collate a taxonomy of values; which then they used to annotate a subset of Anthropic RLHF dataset. On a subset, the inter-rater agreement among the 5 annotators is reported to be high; so then the authors use their labeled dataset to annotate the rest of the Anthropic RLHF dataset. Analysis of both, the subset, and whole dataset are presented; with some instances of toxic or else problematic content being among the chosen preferences in the original dataset.

---

> ### Author Rebuttal · Authors · 2024-08-17
>
> We thank the reviewer for their thoughtful review and valuable feedback on our paper. We appreciate your recognition of the importance and potential impact of our work. We acknowledge the issues raised and are committed to addressing them to improve the quality and clarity of our paper. Here are our responses to the specific points:
>
> 1) Writing and presentation: We will conduct a thorough copy-edit of the entire paper to eliminate typos and ensure consistent formatting.
>
> 2) Inter-annotator agreement calculation: All five annotators independently coded the same set of 200 preferences to calculate this score. For each preference, annotators could choose from seven options corresponding to our seven human value categories. Each preference could be assigned one of our taxonomy's seven human value categories: Information Seeking, Wisdom/Knowledge, Duty & Accountability, Civility & Tolerance, Empathy & Helpfulness, Well-being & Peace, and Justice & Human/Animal Rights.
>
> 3) Value assignment to papers: During our literature review, we focused on identifying a primary or dominant value discussed in each paper. For instance, the paper on Wisdom, expertise, and the application of ethics by Dorothy Nelkin was mapped to the value of Wisdom/Knowledge.
>
> 4) Mutual exclusivity of human values: Thank you for this insightful question. During our annotation process, we encountered instances where multiple values could potentially apply to a single RLHF preference. We identified and assigned the dominant or most salient value to the preference in these cases. This approach was chosen to maintain consistency and to enable clearer quantitative analysis of the dataset. For example, a preference might involve both information seeking and empathy, but one value might be more central to the core intent or outcome of the interaction. In such cases, annotators were guided to select the value that seemed to be the primary driver or goal of the preference. We agree that this approach has limitations, and we will include them in the limitations section of our paper. For instance, in future work, we are considering a multi-label approach to annotation that allows for identifying multiple values per instance. This approach could provide a more nuanced understanding of how different values co-occur and interact within RLHF preferences.
>
> 5) Focus on Western values: Our focus on Western values is largely a result of our literature review process. The journals we selected for our review (as mentioned in Section 3.2) are predominantly Western in their academic tradition and authorship. Additionally, the Anthropic/hh-rlhf dataset we analyzed is likely to reflect primarily Western values, given the context of its creation. Many large language models and their training data have been developed in Western contexts, particularly in the United States. We acknowledge that this Western focus is a limitation of our study.
>
> 6) Regarding your question about how our taxonomy relates to other ethical frameworks, such as Schwartz's Theory of Basic Human Values, our taxonomy is more specifically tailored to the context of AI ethics and RLHF datasets. In contrast, Schwartz's framework is designed for broader cross-cultural comparisons and takes on a more macro outlook.
>
> We believe these changes will significantly improve the quality and clarity of our paper, addressing the concerns raised. We appreciate your detailed feedback. Thank you for your time and expertise in reviewing our work.

---

> > ### Comment · Reviewer_fZFM · 2024-08-27
> > **Review rebuttal**
> >
> > Thank you for the response.
> >
> > I appreciate the comments, though it would have been much better if they also always included how the specific concern will be addressed in the revised manuscript (e.g., a pointer to the section or table).
> > If the response will be made more concrete like that, I might consider a higher mark; for now, I increase it by one.

---

> > > ### Author Rebuttal · Authors · 2024-08-30
> > >
> > > Thank you for your additional feedback and continued engagement with our paper. We have made the following changes to address the feedback, including:
> > >
> > > 1) Writing and presentation: we have conducted copy-editing of the entire paper, focusing on consistent capitalization of value categories, ensuring consistent number formatting throughout the text, correcting plural/singular matches, and fixing typographical errors.
> > >
> > > 2) Inter-annotator agreement calculation: We edited the text in line 182 to specify that annotators were asked to choose from seven options corresponding to our seven human value categories for each preference. The updated text now reads as follows:
> > >
> > > "Before coding all the 6,501 preferences, we calculated an inter-annotator agreement score by having all the annotators independently code the same 200 preferences by choosing one of seven human value categories for each preference. Next, we compared the codes assigned by each annotator to the same preferences to assess the level of agreement or consistency between all the annotators."
> > >
> > > 3) Literature review for value taxonomy: We expanded the content in Section C of the supplementary material to exhaustively detail our taxonomy development process. Specifically, after the current paragraph in that section, we added the following content that describes our process:
> > >
> > > "We employed a multi-stage process to identify, analyze, and categorize relevant literature to ensure a robust foundation for our taxonomy. Our process began with a targeted search of nine ethics-focused journals, including The Journal of Value Inquiry; Axiomathes; The Journal of Ethics; Noûs; Ethics; The Philosophical Review; Science, Technology, & Human Values; Utilitas; and The Journal of Philosophy. Using the keyword "human value," we conducted an unrestricted search across these databases on the last day of October 2023, sorting the results according to relevance. This initial search yielded diverse research materials across the selected journals. The breakdown of search results and articles collected from each journal was as follows: Journal of Value Inquiry (231 out of 1,964 results), Axiomathes (1 out of 1), The Journal of Ethics (175 out of 384), Noûs (269 out of 541), Ethics (400 out of 3,769), The Philosophical Review (240 out of 544), Science, Technology, & Human Values (581 out of 1,744), Utilitas (256 out of 280), and The Journal of Philosophy (208 out of 4,136). We sought to remove duplicates at the source during this data collection process. This process resulted in the collection of 2,361 research materials for analysis.
> > >
> > > Next, we leveraged Rayyan.ai, a collaborative tool for organizing literature for systematic review, to support our analysis. Based on further review of the metadata of the 2,361 articles, including their DOI number, title, publication dates, and authorship information, we identified and removed an additional 35 duplicate articles.
> > >
> > > Following this deduplication process, we applied a four-criteria eligibility criteria to the remaining 2,326 articles. Papers were eligible for inclusion if they met all four criteria, including (1) the full text is accessible either via open access or using our institution's library sign-on credentials, (2) the text is written in English, (3) the text is a research article and not supplemental content (e.g. news report, short book review, newsletter, etc.) (4) the text explores or discusses a core human value or set of values that is relevant to the development, deployment, or examination of the impact of AI systems. By a core set of values in this context, we refer to foundational values such as, but not limited to, Justice, fairness, autonomy, and privacy. By other values, we refer to specific values relevant to AI, including but not limited to transparency, accountability, safety, and human dignity, among others. In addition, papers about values in different contexts, including healthcare, autonomous vehicles, and social media algorithms, were considered. We adopted an interpretivist approach, empowering ourselves to use our informed subjective judgment to determine papers to include, depending on their direct or indirect application to the context of AI. This screening process involved the review of the title, abstract, and content of the specific literature.
> > > Through this process, we excluded any paper that we did not have access to (158), was not written in English (0), is not a research article (425), and did not discuss human values in ways that are meaningfully relevant to examining AI systems (1,623). Above all, this process yielded 120 articles that we used for the creation of the taxonomy.
> > >
> > > Next, we transitioned to developing the taxonomy. Our guiding question was: what is the most dominant human value explored or discussed in this article? Through this approach, we assigned a human value to every shortlisted article.
> > >
> > > We then created the taxonomy through a three-step process. First, we grouped similar values into broader categories, creating a hierarchical structure, like categorizing fairness, rights, and equity under justice/rights. Next, we examined these groupings by ensuring that the specific values logically fit within their overarching categories, like discernment and competence under wisdom and knowledge. Finally, we reviewed to see that values in each group reasonably aligned within the same ethical framework, though not intended to be sacrosanct. Above all, this process allowed us to create the human values taxonomy that supported our classification and audit of the human values embedded within RLHF datasets.
> > >
> > > We make the research articles curated from this process available via this GitHub URL: (https://github.com/ikeobimummc/value_imprint_rlhf/blob/main/human_values_taxonomy/Taxonomy%20Literature/README.md). Above all, this process adopted an interpretivist approach conducted by three researchers who are experienced in mixed methods research and systematic reviews."

---

> > ### Author Rebuttal · Authors · 2024-08-30
> >
> > 4) Relation to other ethical frameworks: We created a new section in the supplementary material (appendix) to briefly discuss the difference between our framework and Schwartz's Theory of Basic Human Values. We specifically added the following text:
> >
> > Comparison with Schwartz Human Values in the Context of AI:
> >
> > The human values taxonomy developed in this paper, while sharing some similarities with Schwartz's Theory of Basic Human Values, presents a framework more specifically tailored to the ethical considerations and operational requirements of AI systems, particularly in the context of auditing the human values embedded within RLHF datasets. Some juxtapositions between both frameworks are highlighted below:
> >
> > a) Contextual Specificity: The values identified in our paper (e.g., Information Seeking, Wisdom/Knowledge, Duty & Accountability) are more directly applicable to AI-human interactions and decision-making processes. In contrast, Schwartz's values (e.g., Self-Direction, Stimulation, Hedonism) are broader and more focused on general human motivations and behaviors.
> >
> > b) Technological Relevance: Our framework includes values like Information Seeking and Wisdom/Knowledge, which are particularly relevant in the context of AI as information processing and knowledge generation systems. Schwartz's theory, developed before the AI era, does not explicitly address these technological aspects.
> >
> > c) Accountability and Transparency: The Duty & Accountability value in our framework is particularly relevant to ongoing discussions about AI transparency and responsibility. This specific focus is absent in Schwartz's more general human-centric value theory.
> >
> > d) Operational Focus: The values in this paper, such as Information Seeking and Duty & Accountability, have a more operational focus, directly applicable to AI functionalities and behaviors. Schwartz's values, while comprehensive for human societies, do not directly translate to actionable AI behaviors or decision-making processes.
> >
> > Hence, while Schwartz's theory provides a framework for understanding human values across cultures, our human value taxonomy offers a more focused, AI-centric approach, especially in examining the human values embedded within AI datasets.
> >
> >
> > 5) Expanding the limitations section: We also expanded Section D of the supplementary material (Potential Limitations of this Approach) and added the following text in line 90 to provide a reason for the Western-focused nature of the values taxonomy, including "are primarily Western-focused because of the Western-centric nature of our literature review sources and the Western-oriented focus of the discourse in the Anthropic/hh-rlhf dataset.
> >
> > Thank you once again. We appreciate your feedback, which has made our work better. We hope the above changes have addressed all the items mentioned in the feedback. Thank you!

---

> > > ### Comment · Reviewer_fZFM · 2024-08-31
> > > **more comments addressed**
> > >
> > > thank you, increasing the mark to 7.

---

> > > > ### Author Rebuttal · Authors · 2024-08-31
> > > >
> > > > Thank you once again! We sincerely appreciate your thoughtful engagement with our work and your decision to increase the score.

---

> > ### Author Response · Authors · 2024-09-04
> > **Score Update**
> >
> > Dear Reviewer fZFM,
> >
> > Thank you once again for your valuable feedback and engagement with our work. You mentioned you would revise your score for our paper to 7, but it appears that this update has not been reflected in the system. Could you please take a look to review and update? Thanks again for all your feedback, time, and expertise.

---

### Official Review · Reviewer_rTbf · 2024-07-23
**Useful contribution on the whole**

**Rating:** 6
**Confidence:** 4
**Correctness:** For the most part yes, with the excep…
**Clarity:** Yes with the exception of W1.

**Review:**

See below.

**Strengths:**

S1) The proposed approach appears useful for classifying values in RLHF datasets.
S2) The three-stage process is well-motivated. It involves developing a taxonomy, manually annotating over 6000 samples using the taxonomy, and training models to evaluate larger datasets.

**Additional Feedback:**

..

**Documentation:**

..

**Limitations:**

..

**Opportunities For Improvement:**

W1: The “ethical paradigm” aspect of the taxonomy is problematic. This should be removed.

The authors do not provide an explicit discussion or justification of the “ethical paradigm” component of their taxonomy, nor of specifically how they applied it to the data.

The ethical paradigms listed (deontology, virtue ethics, and utilitarianism) have uneven granularity. Deontology and virtue ethics are broad families of moral theories, whereas utilitarianism is a specific subtype of consequentialism. This inconsistent granularity is misleading in the context of a taxonomy.

The association of ethical paradigms with mutually exclusive values is misleading. Deontology, virtue ethics, and consequentialism can all accommodate the full list of human values included in the taxonomy. Annotating RLHF preferences based on these paradigms would require a different approach than simply associating these paradigms with mutually exclusive values.

**Relation To Prior Work:**

Yes.

**Summary And Contributions:**

Authors propose a method for classifying the values embedded in RLFH datasets, involving 1) a taxonomy of values, 2) a manual qualitative annotation process using the taxonomy, 3) a process for using (1) and (2) to train transformer-based models that can be used to audit larger RLHF datasets.

---

> ### Author Rebuttal · Authors · 2024-08-17
>
> We thank the reviewer for their thoughtful review and feedback on our paper. We appreciate your recognition of the usefulness and well-motivated nature of our proposed approach for classifying values in RLHF datasets.
>
> Regarding the "ethical paradigm" aspect of our taxonomy (W1), we acknowledge the validity of your points and appreciate the opportunity to address them. After careful consideration of your suggestion, we are making the following revision to the paper:
>
> 1) Remove the explicit "ethical paradigm" categorization: We agree that the current presentation of ethical paradigms may introduce unnecessary complexity around ethical categorization of values. We will remove this categorization from our primary analysis. Instead, we will concentrate our analysis solely on the human values identified in the RLHF datasets, which form the core contribution of our work.
>
> By implementing this change, we believe we can maintain the focus of our research on our primary goal which is to provide a robust method for classifying the human values embedded in RLHF datasets, allowing us to present our findings more clearly.
>
> We appreciate your feedback, as it has helped us improve and tighten our work. Thank you for your time and expertise in reviewing our paper.

---

> > ### Comment · Reviewer_rTbf · 2024-08-26
> > **Thank you**
> >
> > Thank you for engaging with the feedback!

---

### Official Review · Reviewer_zqWr · 2024-07-26
**The paper provides a useful method of analysis of values in RLHF datasets that is backed by a robust methodology.**

**Rating:** 8
**Confidence:** 5
**Clarity:** As stated above, the paper is well wr…

**Review:**

The contributions are clearly stated and well-motivated. The introduction very nicely summarized what was to come. The authors also make clear how their paper is distinct from prior work. The work is well-documented. I appreciate the inclusion of a data sheet in the appendix. The contributions may be a bit more limited than is typical of the track, but I believe the strong motivation and situatedness of the work improves the integrity of its contributions.

**Strengths:**

The work is significant for the broader community focused on alignment and provides a robust methodology for assessing values in RLHF datasets. I also commend the authors on providing not only a well-written, well-situated, and compelling paper, but also for their attention to thoughtfully discussing limitations and providing thorough documentation in their appendices.

**Additional Feedback:**

N/A

**Correctness:**

The findings are well in line with the research questions scoped and the conclusions are supported by the data produced in the study.

**Documentation:**

The dataset is well-documented, using a datasheet to provide details.

**Limitations:**

Limitations are well-discussed, including limitations specific to the qualitative coding process described in the appendix.

**Opportunities For Improvement:**

The authors might consider creating a table with the codebook definitions. The in-text descriptions of how similar codes differed from each other were helpful, but such a table would be handy as a reference for quickly parsing other tables and results.

Even though the authors made an effort to quantify the ethical paradigms represented in the dataset, they do not discuss these results at all. The results raise questions for me regarding ethical and moral motivations in technology development and machine learning as well as the extent to which these results reflect discussions in other work such as Birhane et al. (2022). More generally, some additional unpacking of the implications of these findings would improve the impact of these findings (space permitting). Personally, this discussion could take the place of section 5.2. The finding that unethical preferences were overlooked in these datasets is an important check on quality, but, in my opinion, less central to the research questions than a discussion of the roots of value imbalances in RLHF datasets.

Typo? “The proposed approaches that researchers could 87 adopt to mitigate these biases.” (Beginning of the second paragraph of the section 2.1)

**Relation To Prior Work:**

The work very clearly distinguishes itself from prior work.

**Summary And Contributions:**

The authors contribute an analysis of values embedded in an RLHF dataset using a solid approach that can be reused and adapted to analyze other datasets. They also contribute an accurate modeling approach for scaling their analysis.

---

> ### Author Rebuttal · Authors · 2024-08-17
>
> We thank the reviewer for their thoughtful review and feedback on our paper. We appreciate the positive assessment of our work's contributions, methodology, and overall presentation. We are glad the reviewer found our paper well-documented and situated within the broader context of AI alignment research.
>
> Response to suggestions for improvement:
>
> 1) Codebook table: We agree that including a table with codebook within the main section of the paper would enhance the ease of parsing through the paper. We will add an abridged codebook of the taxonomy to section 3 that summarizes the values and their definitions in a way that complements the in-text descriptions in section 4.
>
> 2) Implications: We agree that a more in-depth discussion of the implications of our findings would strengthen the paper. We will revise section 5.2 to focus more on unpacking the significance of the value imbalances we observed in RLHF datasets. Specifically, we will discuss how values embedded within RLHF datasets are an affordance in the context of LLMs and, through that lens, discuss the implications of the value imbalances.
>
> 3) Typo: Thank you for catching this error. We will correct the sentence so it reads more clearly.
>
> 4) Ethical paradigm: Upon careful consideration of all reviewer feedback and re-evaluation of our research focus, we have decided to streamline our analysis to concentrate primarily on the human values embedded in the RLHF datasets and moving discussions about ethical paradigm to section 5.1. We believe this approach will strengthen the paper's focus and impact while still addressing the important ethical dimensions of our work.
>
> We appreciate the reviewer's constructive feedback and will incorporate these suggestions to improve the paper's clarity, depth, and impact. Thank you again for your thorough review and helpful comments.

---

### Official Review · Reviewer_rChu · 2024-07-29
**Value Imprint: A Technique for Auditing the Human Values Embedded in RLHF Datasets**

**Rating:** 3
**Confidence:** 4
**Correctness:** Yes
**Clarity:** See Review

**Review:**

Pros:
The taxonomy of human value is interesting.

Cons:
1. The main contribution of this paper is a framework for auditing and classifying the human value in RLHF datasets. However, this framework is only tested on a single dataset. More case studies are needed;
2. This framework involves only manual labeling and classifier training, which is somewhat trivial.
3. Section 3 is in lack of running examples for illustration purposes.

**Strengths:**

See Review

**Additional Feedback:**

none

**Documentation:**

Yes

**Limitations:**

Yes

**Opportunities For Improvement:**

Can existing LLMs be used to create the datasets in Section 3 for training?

**Relation To Prior Work:**

Yes

**Summary And Contributions:**

This paper introduces a framework for auditing and classifying the human values embedded in RLHF datasets. This paper conducts a case study with two phases. In the first phase, a taxonomy of human values is developed and a dataset is constructed. In the second phase, a classifier is trained to study the most dominant human values within the dataset.

---

> ### Author Rebuttal · Authors · 2024-08-17
>
> We thank the reviewer for an insightful review of our paper and appreciate the opportunity to address the points raised. We agree that testing our framework on additional datasets would help strengthen its generalizability. To address this and other suggestions, we will do the following:
>
> 1) We will expand our analysis to include one major RLHF dataset to demonstrate the generalizability of our framework.
>
> 2) We will also include a section in the appendix that discusses generalizability and how our framework can be adapted to classify human values in different RLHF datasets.
>
> 3) While the core components of manual labeling and classifier training may appear straightforward, we believe the novelty and contribution lies in what our framework enables within the LLM development process. Specifically, our work provides a pathway that enables researchers and AI practitioners to audit and examine the human values embedded within RLHF datasets before integrating them to their pipeline. This process is currently lacking within the LLM development process and cycle.
> 4) We appreciate the suggestion to include running examples in Section 3 and will incorporate illustrative examples for each step of our methodology.
>
> Overall, we appreciate the reviewer’s recognition of the interesting nature of our human value taxonomy and the overall contribution of our work. Your feedback has helped us strengthen the paper and increase its potential for impact.

---

> > ### Author Rebuttal · Authors · 2024-08-31
> >
> > Thank you once again for your feedback. We have added two case studies to Section F of the Appendix/Supplementary material section of our manuscript. The case studies were conducted on the **OpenAI WebGPT Comparisons** dataset and the **Instruction Tuning with GPT-4** dataset created by researchers from Microsoft. Both datasets were generated using LLMs and human annotation. We selected these datasets due to their recent relevance within the RLHF/AI Alignment community, as shown by the number of downloads and citations received. We briefly describe our case study setup for each dataset and have provided in-depth information in the manuscript.
> >
> >
> > ### Case Study 2: OpenAI WebGPT Comparisons Dataset
> >
> > We collected the OpenAI WebGPT Comparisons dataset from the Hugging Face library and focused our case study analysis on the content of the `question` and `answer_0` columns. We created a function to extract only the `full_text` from the `question` column and dropped the non-essential metadata, including `"triviaqa"`, `"dataset"`, and `"id"`. Next, we concatenated the content of the updated `question` and `answer_0` columns into a new column to form a complete preference unit. We then used our model to classify these preferences to examine the human values embedded within them. Our analysis revealed that **Wisdom/Knowledge (78.17%)** was the most common human value in the dataset. More results on this case study are provided in the supplementary section of our manuscript.
> >
> >
> > ### Case Study 3: Alpaca GPT-4 Dataset
> >
> > We fetched the Alpaca GPT-4 dataset from the GitHub repository dedicated to the project. Next, we concatenated the `instruction` and `output` columns from the original dataset into a new `combined` column, creating a complete human preference conversation. We reduced the DataFrame to contain only this new `combined` column and then conducted our case study classification analysis. Findings from our analysis showed that **Wisdom/Knowledge (66.56%)** was the most common value. We have added more details about the results to our manuscript.
> >
> >
> > ### Running Examples
> >
> > We have added running examples throughout Section 3 to support readability.
> >
> >
> > ### Can LLMs be Used to Create a Human Values Dataset?
> >
> > This is an insightful question. The current generation of LLMs can generate synthetic human value datasets. However, there are benefits and limitations to this approach. The obvious benefit involves the speed and scale at which these datasets can be created. However, the limitation is that when using LLMs to generate synthetic human value datasets, extensive human validation would be necessary to ensure accuracy, relevance, and alignment with the given society's values. Another limitation includes concern about hallucinations. Hence, while this approach is technically possible, limitations might impact its efficacy at scale.

---

> > ### Author Rebuttal · Authors · 2024-08-31
> >
> > # Human Values Embedded within OpenAI WebGPT and Alpaca GPT-4-LLM Datasets
> >
> > | Human Values                    | OpenAI WebGPT | Alpaca GPT-4-LLM |
> > |:------------------------------|:------------------------------------:|:---------------------------------:|
> > | Civility/Tolerance             | 1.46%                                | 1.05%                              |
> > | Duty/Accountability            | 1.39%                                | 0.18%                              |
> > | Empathy/Helpfulness            | 0.90%                                | 2.92%                              |
> > | Information Seeking            | 5.67%                                | 26.45%                             |
> > | Justice & Human/Animal Rights  | 0.04%                                | 0.17%                              |
> > | Well-being/Peace               | 12.37%                               | 2.67%                              |
> > | Wisdom/Knowledge               | 78.17%                               | 66.56%                             |
> >
> >
> > > Datasets containing the human values classification results of OpenAI WebGPT and Alpaca GPT-4-LLM preferences are now on GitHub.
> >
> >
> > Thank you once again for your feedback which has made our work better.

---

### Author Response · Authors · 2024-09-01
**Summary of Response**

We thank all the reviewers for their comments and valuable feedback. In response to your collective feedback, we have made several revisions to strengthen our paper:

1. **Expanded Analysis:**
We have expanded our analysis to include two RLHF-related datasets (OpenAI WebGPT Comparisons and Alpaca GPT-4-LLM) to demonstrate the generalizability of our framework. Details of these case studies have been added to the Appendix.

2. **Revised Sections 5.1 and 5.2:**
We have revised Sections 5.1 and 5.2 to focus on unpacking the significance and implications of the value imbalances we observed in RLHF datasets through the lens of human values as affordance and the connection between ethical paradigms and the need for human value benchmarks.

3. **Enhanced Taxonomy Development Description:**
We have expanded our description of the taxonomy development process, providing more details on our literature review methodology and the process of assigning values to papers.

4. **Edited Human Values Taxonomy Table:**
We have edited our human values taxonomy table to focus more clearly on the human values embedded in the RLHF datasets.

5. **Thorough Copy-Edit:**
We have conducted a thorough copy-edit of the entire paper to eliminate typos, ensure consistent formatting, and improve overall clarity and readability.

6. **Comparison to Schwartz's Theory of Basic Human Values:**
We have included a new section in the Appendix comparing our framework to Schwartz's Theory of Basic Human Values, highlighting the AI-specific focus of our taxonomy.

7. **Expanded Limitations Section:**
We have expanded the limitations section to address the Western-focused nature of our values taxonomy.

These revisions, along with others mentioned in the different threads, have significantly strengthened our paper, addressing the concerns raised while maintaining the core contributions of our work.

Once again, we sincerely thank all reviewers for their time and valuable feedback.

---

### Decision · Program_Chairs · 2024-09-26

**Decision:**

Accept (Spotlight)

**Comment:**

Meta review of Value Imprint: A Technique for Auditing the Human Values Embedded in RLHF Datasets
NeurIPS 2024 Datasets and Benchmarks Track Submission 2336

The topic of how RLFH improves AI in the fine-tuning process has been taken largely as a given in mainstream discussion, as a net positive way to imbue models with human preferences. How this actually happens and, on aggregate, what values this process imbues, has been under-discussed in the literature. The authors position their work well within the broader literature on values imprinted in AI and LLM systems, pointing out the relevant papers focus on the systems themselves, and not on the distinct process of RLHF.

Most of the reviewers agree that this paper constitutes a valuable contribution to the field. However, one weakness that Reviewer rTbf points out is the authors’ flawed approach to ethical paradigms, with “uneven granularity” between the categories defined by the authors. I agree with this assessment, and appreciate the authors revisions in changing the label of the approach.

The lowest rating was from Reviewer rChu, who pointed out that a weakness of the paper is a lack of generalizability, as the paper tested only 1 dataset. The authors responded by adding an additional 2 RLHF datasets to their case studies, which would address that concern for a future iteration of the work. The reviewer did not respond to this rebuttal. Because this improvement is so substantial, I believe it would have merited an improved score from the reviewer, had they responded. At the same time, additional results are only permitted during rebuttal to clarify concerns; the decision to accept a paper must be based on the original submission.

For its contribution to the state of the art, potential to provoke discussion, and sufficient response to the reviewers’ stated weaknesses in the paper, I argue that this paper should be accepted nevertheless as a spotlight poster.